# Position: 'AI Alignment' Encompasses Competing Technical Priorities

**Tushita Jha** [* 1]   **Rory Svarc** [* 2]   **Mateusz Bagiński** [3]

## Abstract

The ML literature contains many distinct concepts falling under the heading of 'AI alignment'. After noting three concepts of AI alignment in the context of their corresponding research programs, we claim that realistic interventions may promote 'AI alignment' under one conception while being actively counterproductive from the perspective of others. We suggest that tensions between alignment ideals emerge due to differences in *background threat-models*, alongside differences in *normative* orientations. In light of our analysis, researchers aiming to further the goal of 'AI alignment' should do five things. First, they should not conflate distinctions of policy and distinctions of scientific scope; second, methodological disagreements should be acknowledged explicitly; third, researchers should distinguish between 'AI alignment' as a high-level ideal and specific 'alignment proxies' used in empirical research; fourth, they should use more granular concepts to identify both the *source* and *nature* of possible AI harms/benefits; fifth, they should explicitly acknowledge the diversity of 'alignment' concepts in both empirical work and in communication with non-technical audiences.

## 1. Introduction

'Alignment' is a binary relation. To say that $x$ is aligned with $y$ is, in natural parlance, to say that $x$ and $y$ stand in some position of agreement or congruity. When we speak of 'AI alignment', two questions must therefore be answered:

Q1. What are the target properties $y$ that $x$ must satisfy?

Q2. What is the $x$ that must satisfy target properties $y$?

---

[*]Equal contribution   [1]Mimir Center for Long Term Futures Research, Sweden [2]Arb Research, Prague 11636, Czech Republic [3]AFFINE. Correspondence to: Tushita Jha <particle.mania@gmail.com>, Rory Svarc <rory.svarc@cantab.net>.

*Proceedings of the 43rd International Conference on Machine Learning*, Seoul, South Korea. PMLR 306, 2026. Copyright 2026 by the author(s).

One central contention of this paper is that answers to such questions are more difficult – and indeed more fraught – than they might initially appear. Many papers give passing definitions of 'AI alignment': as the task of ensuring AI systems "adhere to or act in accordance with human values" (Sucholutsky & Griffiths, 2023; Yeh et al., 2024), or simply the "intended goals, interests, and values, as envisioned by their designers" (Li et al., 2023). The earliest uses of the term were in the context of managing the risks and opportunities of highly advanced AI systems of the future (Soares & Fallenstein, 2014; Russell, 2015; Yudkowsky, 2016; Taylor et al., 2016; Future of Life Institute, 2017). Other definitions have since framed AI alignment as requiring that "application developers" can tune models "according to the social norms and values of their user community" (Varshney, 2024), or that AI systems possess 'goals' that are "aligned with the goals of humans" (Ngo et al., 2025; Gao et al., 2023). Many more specific concepts of 'AI alignment' have likewise emerged: from Thick (Foster, 2023) to Collective (The Collective Intelligence Project, 2023) to Socioaffective (Kirk et al., 2025) to Decolonial (Varshney, 2024) forms of 'alignment'.

We read these distinct formulations of 'AI alignment' as distinct answers to {Q1, Q2}. The paper hence begins by outlining three high-level ideals falling under the heading of 'AI alignment', each comprising a different set of answers to Q1 and Q2 (Section 2), and consequently a different set of implicit properties required for 'aligning' $x$ to $y$. Thereafter, Section 3 introduces two distinctions which cut across the three alignment concepts heretofore introduced. In turn, these cross-cutting distinctions are used to illustrate how existing ideals for 'AI alignment' may be *in tension* with one another. Sections 2 and 3 therefore comprise the primary argument for the paper's position. Per Section 2, the term 'AI alignment' is polysemous. Moreover (per Section 3) this polysemy obscures normative disagreements behind ostensibly 'technical' conceptions of AI alignment.

We then move to the implications of our analysis in Section 4. First, we suggest that distinctions of *scientific or technical scope* should not be conflated with policy distinctions. Second, we suggest that the methodological sources of potential disputes over 'AI alignment' should be acknowledged more explicitly. Third, we suggest that reviewer pooling and conferences based on different alignment ideals should

be instantiated. Fourth, we argue that 'proxy' alignment concepts should be introduced where appropriate. And, finally, we argue that the diversity of different 'alignment' concepts should be both explicitly acknowledged and communicated appropriately to policymakers and non-technical audiences. Section 5 responds to alternative views; Section 6 concludes.

## 2. Concepts of Alignment

This section outlines distinct uses of 'AI alignment' in the context of the two questions outlined in the introduction: (i) 'what are the target properties $y$ that $x$ must satisfy?', and (ii) 'what is the object $x$ that ought to satisfy $y$?'. We suggest that operative high-level conceptions of AI alignment often disagree about *what we're trying to align*, rather than simply disagreeing about *the target properties* for AI alignment.[1]

### 2.1. The Prosaic View of Capable AI

'AI alignment' involves 'aligning' systems which are 'artificially intelligent'. Hence, we might ask what is being invoked when we talk about 'intelligence' in the context of AI systems. In its most minimalist form, we might say that the concept 'intelligence' is meaningful only insofar as we can *use* that system in order to achieve some task competently, effectively, and reliably. Hence we would say that a model trained to play Atari is (more) 'intelligent' insofar as it (more) *effectively plays Atari*; likewise, a large reasoning model trained to solve mathematical tasks is more intelligent the more successfully it solves novel mathematical tasks.

Correspondingly, our first concept of alignment says that AI systems are 'aligned' when they reliably execute tasks given to them by the human users. We may 'align' a large language model to follow instructions (Si et al., 2025), align a vision model to produce accurate textual descriptions of images (Liu et al., 2024a; Yarom et al., 2023), or align a language model by reducing its tendency to produce 'hallucinations' or confabulations (Chen et al., 2024). Under this conception of alignment, "aligning" AI systems simply comes down to improving their task-execution capacity.

**Definition 2.1** (Task Reliability). An AI is 'aligned' if it does that which we want it to do. Hence, a system is *misaligned* insofar as its outputs fall short of the maximally

competent outputs in at least one respect.

Task Reliability has taken on a more specific guise after the advent of user-facing LLMs, found perhaps most notably in OpenAI's original InstructGPT paper. It is claimed that: "[LLMs] can generate outputs that are untruthful, toxic, or simply not helpful to the user. In other words, these models are *not aligned* with their users" (Ouyang et al., 2022). We may call this conception *Alignment as Fine-Tuning*, where an LLM is 'aligned' if its observable behaviors adhere to the desired behaviors of the AI's users (or indeed developers). Note however that Alignment as Fine-Tuning can be seen as a subtype of Task Reliability. In both cases the object we're trying to align ($x$) are the task-specific capabilities of an AI system in some sense, and the set of desired target properties ($y$) is defined by the intentions of its developers. Hence, we subsume all discussions of 'Alignment as Fine-Tuning' under the heading of 'Task Reliability' moving forward.

In the context of aligning an artificial 'generally intelligent' system (AGI), an AGI can be considered aligned to the extent that the AI system's performance on narrow tasks composes into the capacity to execute multiple tasks in a coordinated manner when this is instrumental towards serving some specific goal. Indeed, understandings of 'intelligence' as *general competence* are referenced both in early discussions of AI research ('intelligence' measures "an agent's ability to achieve goals in a wide range of environments" (Legg & Hutter, 2005)), and in the reports and publications of leading AI labs. DeepMind's classificatory 'levels of AGI', for example, involves defining a 'competent AGI' as a system that achieves equivalent performance to the "50th percentile of skilled adults" on a "wide range of non-physical tasks, including metacognitive tasks like learning new skills" (Morris et al., 2024). Similarly, OpenAI's charter defines 'AGI' as "highly autonomous systems that outperform humans at most economically valuable work" (OpenAI, n.d.).

### 2.2. Alignment, Biases, and Social Situatedness

An alternative conception of alignment is more closely linked to conceptions of the broader social good – and the relevance of the sociotechnical functions occupied by AI systems. Example concerns include AI systems contributing to misinformation (Neumann et al., 2022; Longoni et al., 2022; Gehrmann et al., 2025), exacerbating inequality and oppression (Bender et al., 2021; Prabhakaran et al., 2022; Lee et al., 2024; Stoev et al., 2023), or recommender systems contributing to addiction or polarization (Stray et al., 2021; Agarwal et al., 2024; Prunkl, 2024; Smith et al., 2024). Even if (for example) recommender systems in some sense function in accordance with their designers' wishes (e.g., by maximizing clickthrough rate), such systems are sometimes

---

[1]The order in which this section introduces its three concepts admits some chronological laxity. Although the first concept we introduce bears close resemblance to more demotic uses of the word 'alignment', the noun phrase 'AI alignment' first appears in the context of research concerning our *third* alignment concept around 2016 (Yudkowsky, 2016; Taylor et al., 2016; Christiano, 2016), with later work by academics and industry researchers picking up the phrase in subsequent years (Irving et al., 2018; Hadfield-Menell & Hadfield, 2018; Vamplew et al., 2018), resulting in the more widespread and polysemous use of 'AI alignment' seen today.

claimed to be *misaligned* with broader social values such as cohesion or truthfulness (Zhuang & Hadfield-Menell, 2020).

One can observe similar critiques about LLMs. One may claim that the widespread use of LLMs will result in undesirable 'homogenization effects' (Moon et al., 2025) or create echo chambers (Sharma et al., 2024; Nehring et al., 2024). Or perhaps one thinks that LLMs will cause epistemic harms, either through expressing 'opinions' at odds with the broader population (Santurkar et al., 2023; Liu et al., 2024b) or as a result of LLMs' 'woke biases' (Oremus, 2023). In its most general form, we can hence define the following alignment concept.

**Definition 2.2** (Social Judiciousness)**.** An AI system is 'misaligned' if the system's outputs – given the contexts in which it is deployed – *create*, *perpetuate*, or *exacerbate* undesirable societal trends.

Note that Social Judiciousness is distinct from Task Reliability in at least two ways. First, the complaints listed are not necessarily claims about 'incompetence' on part of developers; such failures may arise due to developers' normative blindspots, because unreliable/damaging AI systems are deployed with insufficient caution, or indeed because of self-consciously nefarious motives. This constitutes a difference in the metric $y$ against which 'AI alignment' should be judged. Second, many of the complaints listed construe 'AI systems' as '*sociotechnical*' systems. Evaluating the 'alignment' of AI systems involves looking not only at the model as technical artifact, but also in the context of "social relations" (Johnson & Verdicchio, 2025), beyond "individual technical artifacts" like "data, model architecture, and sampling" (Weidinger et al., 2023). For this reason, Social Judiciousness diverges from Task Reliability by rejecting the assumption that the $x$ we're trying to align to some external standard is a *technical artifact*—viz., 'the AI system'.

The significance of this framing surfaces in the context of claims about deleterious trends being easier to amplify and sociotechnically reproduce, while judicious design and use require *deliberate* and *responsible* intent. In this sense, the societal trends relevant for examination here (i.e., those with systemic scope) might appear as default direction for ordinary AI progress, and this brings forth two possible *sources* of failure from the perspective of 'Social Judiciousness'. In the first instance, sources of harm may enter AI systems due to biases inherent in the data upon which it is trained (and hence reproduces in its output behavior); this might involve training data which is demographically unrepresentative (Buolamwini & Gebru, 2018) or biased in favor of certain ideologies (Moayeri et al., 2024). We call this source of harm *Training Data Conservatism*.

**Definition 2.3** (Training Data Conservatism)**.** Undesirable model behaviors arising from the model's training data being *biased*, *unrepresentative*, or *unreliable*.

However, Social Judiciousness concerns also involve worries about highly persuasive LLMs being successfully used as tools of persuasion (Dehnert & Mongeau, 2022), recommender systems being used by powerful actors to shape the preferences of the less powerful (Laitinen & Sahlgren, 2021; Prunkl, 2024), or (more generally) the use of 'manipulative designs' in human-centered computing (Babaei & Vassileva, 2024). We dub this source of concern about Social Judiciousness a concern of *Malicious Use*.

**Definition 2.4** (Malicious Use)**.** Undesirable model behaviors arising from powerful or malicious actors using AI systems to accomplish their chosen ends.

Although such sociotechnical concerns are not always raised under the explicit heading of 'AI alignment', these diffuse concerns have found explicit conceptualization and crystallization under discussions of 'thick alignment' (Foster, 2023; Nelson, 2023).

## 2.3. Alignment as 'Takeover Avoidance'

An altogether quite different conception of alignment emerges from discussions of potential catastrophic risks resulting from the development of future AI systems, often dubbed 'AGIs' or 'ASIs' (artificial superintelligences). Related concerns about "thinking machines" able to "outstrip our feeble powers" and "take control" date back at least to Turing in 1951 (Turing, 2004), with scattered attention in following decades (Good, 1966; AI World Society, 2021); its more recent development, however, can be traced back to work in the early 2000s and 2010s (Yudkowsky, 2004; 2008; Bostrom, 2014; Russell, 2019; Christian, 2020).

One way to frame this concern focuses on the dangers of AI systems 'optimizing' for, or *bringing about*, some target state of affairs in the real world. In line with what has been termed the "standard view of intelligence" (Russell, 2019), this target is often understood in terms of 'goals' or 'objectives' (Sutton, 2004; Russell & Norvig, 2010, p.5). Importantly, these 'goals' or 'objectives' need not be identical to the intended or specified targets (Hubinger et al., 2021). On this view one may identify 'intelligence' with 'optimization', and believe that the outcomes produced by 'optimization processes' are worthy of distinct consideration from outcomes produced by 'mere accidents' (or mistakes, or random errors).

Alongside other suggestive arguments, the ability of optimization processes to enhance their own capacity to produce optimizing effects (related to what is known as instrumental convergence) — alongside the difficulty of capturing the rich character of human-compatible goals in the AI's optimization target (Yudkowsky, 2009; Armstrong, 2013; Christiano, 2019) — motivates concerns about the *content* of future AI systems' objectives being 'unfriendly' to hu-

mans. Because (so the argument goes) future AI systems will be smarter than humans, we stand in an *adversarial* relationship to these future systems. This is to say that future AI systems will—*qua* optimizing agents—have an incentive to hide their 'true objective' from humans until they are able to pursue their objectives without threat of human interference. Scenarios of this kind—whereby AI systems possess 'long-term' or 'broadly-scoped' objectives that they try to hide from humans—are often referred to as "deceptive alignment" in the wider literature (Carlsmith, 2023; Ngo et al., 2025; Ji et al., 2025). In turn, we can now introduce the concept of *Takeover Avoidance*.

**Definition 2.5** (Takeover Avoidance). An AI system is 'misaligned' if the model optimizes for undesirable effects in the real world.

Researchers motivated by the ideal of Takeover Avoidance often use proxy concepts in the context of empirical work on contemporary systems. Much research has been motivated by '*preferentist*' assumptions, where 'preferences' are considered an adequate representation of optimization targets, with a close connection assumed between rationality and maximizing preference satisfaction (Zhi-Xuan et al., 2025). Hence, some discussions within this tradition make reference to psychologistic language like 'goals' rather than 'optimization' as such (di Langosco et al., 2022; Shah et al., 2022; Bengio, 2023; Kenton et al., 2024; Greenblatt et al., 2024; Betley et al., 2025; Ngo et al., 2025; Meinke et al., 2025).

### 2.4. Situating The Alignment Ideals

This section has introduced three alignment ideals: Task Reliability, Social Judiciousness, and Takeover Avoidance. These high-level ideals represent different answers to Q1 and Q2 raised in the introduction (see Table 1). Although we believe that certain more specific concepts fall under the taxonomy we've outlined here — for instance, we believe that Thick (Foster, 2023), Socioaffective (Kirk et al., 2025), and Decolonial Alignment (Varshney, 2024) are all well subsumed as subtypes under our discussion of Social Judiciousness — the cases of (for example) Collective (The Collective Intelligence Project, 2023) and Bidirectional Alignment (Shen et al., 2025) appear arguably distinct.[2]

Acknowledging that our discussion falls short of exhaustive coverage, the remainder of this paper focuses on just the three concepts of Table 1. More specifically, we shall claim that these concepts represent ideals which in some cases cannot be jointly pursued, and hence that 'AI alignment' encompasses *competing* rather than merely *different* technical

---

[2]The latter two cases, specifically, are more naturally framed in terms of *positive* targets compared to Social Judiciousness's focus on negative societal trends. Section 3.2 provides related discussion on this point.

| Alignment Ideal | What's Being Aligned? | Aligned to What? |
| --- | --- | --- |
| **Non-Takeover** | Optimization Target of AGIs/ASIs | Non-Takeover Targets |
| **Social** | Deployed AIs in Realistic Settings | Some External Normative Standard |
| **Task** | Locally Measurable AI Behaviors | Developer Intentions |

*Table 1.* Answers to Q1 and Q2

ideals.

## 3. Practical Tradeoffs Between Alignment Ideals

In this section we introduce two distinctions with the aim of highlighting tradeoffs between the high-level alignment ideals introduced in Section 2. Section 3.1 and Section 3.2 each begin with a proposition, followed by definitions and arguments aimed at establishing each proposition. We then summarize these tradeoffs in Section 3.3.

### 3.1. Harms from Misdirected Competence vs Harms from Incompetence

**Proposition 3.1.** *Different threat-models lead to tradeoffs between different alignment ideals.*

We will say that a **threat-model class** (hereafter 'threat-model') is characterized by the source or origin of potential negative outcomes that may arise from developing AI systems, such that appropriate interventions can be used to minimize the chance that this set of outcomes occurs. To understand alignment ideals more precisely we must distinguish between two high-level threat models, the first of which we call *Harms from Competence*.

**Definition 3.2** (Harms from Misdirected Competence). *Harms from Misdirected Competence* are threat-models which posit dangers arising from AI systems which are highly *competent* on some set of tasks.

Recall that Takeover Avoidance is motivated by the idea that highly powerful AI systems will, in the future, 'optimize for' or 'pursue goals' contrary to human interests, and thereafter remain impervious to human control. Hence, the threat-model underlying the ideal of Takeover Avoidance involves AI systems in some sense being *too capable*. It is for this reason that the motivating threat-model behind Takeover Avoidance can be classified as *Competence Harm*.

The case of Social Judiciousness is less straightforward, as failures in this case may arise either from AI competence or AI incompetence. While concerns about Takeover Avoidance are *necessarily* concerns about AI competence, researchers focused on Social Judiciousness in many cases

deal with harms resulting from *incompetence*. Consider, for example, criticisms made of the application of ML in predictive policing (Akpinar et al., 2021; Ensign et al., 2018; Fussell, 2020) as a result of models learning shallow or biased associations from their training data, or because they are given 'practically impossible' tasks (Raji et al., 2022). Here, the operative threat-model can be seen — given the unjust society in which we live — as a form of Training Data Conservatism yet to be overcome, and hence an *Incompetence Harm*.

**Definition 3.3** (Harms from Incompetence). Threat-models which posit dangers that arise from AI systems which are *incompetent* on some set of tasks.

We introduce these threat-models as they highlight a source of potential tension between researchers focused on Social Judiciousness and those focused on Takeover Avoidance. Alongside predictive policing, many other cases can be found where researchers raise concerns about socially deleterious effects of AI simply replicating biases in their training data, affecting healthcare (Agrawal & Prabakaran, 2020; Nagendran et al., 2020), facial recognition (Buolamwini & Gebru, 2018), the spread of misinformation, or harms caused by LLMs expressing 'views' which fail to be representative of the broader population (Liu et al., 2024b). For this reason, researchers focused on failures of Social Judiciousness due to *Incompetence Harms* may favor research programs aiming to reduce the tendency of present-day LLMs to 'hallucinate', as this may in turn reduce harms caused by LLMs repeating misinformation found in their training data.

By contrast, those focused on Takeover Avoidance may object to such proposals *precisely because* they make AI systems more competent. From the perspective of Takeover Avoidance, reduced hallucination rates may be undesirable unless they come alongside 'specific countermeasures' (Cotra, 2022); greater degrees of situational awareness may enable misaligned AIs to more effectively 'scheme' against the oversight mechanism (Carlsmith, 2023; Meinke et al., 2025), or increase models' ability to 'sandbag' on certain tasks when being evaluated (Anthropic, 2025, pp. 90-100). This is our first example indicating potential tensions between the ideals expressed by different conceptions of 'AI alignment'. (This tension was noted briefly in an informal precursor to the taxonomy herein (Grietzer & Jha, 2024)).

### 3.2. Positive and Negative Alignment

**Proposition 3.4.** *When evaluating model behavior/outputs, the* scope *of behavioral/output evaluation leads to tensions between different alignment ideals.*

Another source of potential tension between alignment ideals can be illustrated with a cross-cutting distinction. Insofar as Harms from Competence and Harms from Incompetence both specify *threat-models*, they are primarily focused on the potential harms resulting from AI systems. However, we could imagine focusing more specifically on what *positive* properties we would like AI systems to possess. Let us say that 'Positive Alignment' prescribes the set of properties we want AI systems to possess, and 'Negative Alignment' prescribes properties we do *not* want AI systems to possess.

Of course, one may think that talk of 'avoiding undesirable properties' is *just the same thing* as 'producing positive properties'. Indeed, under natural formalizations in first-order logic they would simply express the same concept.[3] However, the distinction between Positive and Negative Alignment arises when the implicit domain of evaluation for the AI's outputs or 'behaviors' differs depending on whether we are checking whether it does what we *want*, or whether the AI *avoids* doing what we *don't want*. In practice, it will often be more straightforward to evaluate whether an AI system performs some fixed task reliably or effectively than to evaluate whether its outputs avoid all of the multifarious ways an AI could falter. For instance, we may train a model to achieve better scores on certain mathematical benchmarks, and concomitantly find that the model has higher 'hallucination rates' than previous SOTA models.[4] This would be progress towards the ideal of *Positive Alignment*, but regress towards the ideal of *Negative Alignment*.

The distinction between Positive and Negative Alignment is particularly important for understanding the potential tensions between Task Reliability and the remaining alignment ideals. When specifying desirable AI behavior, one cannot plausibly list every behavior that the developer does *not* wish the AI to exhibit. However, Social Judiciousness and Takeover Avoidance are concerned with the unintended consequences of deploying AI systems that narrowly behave (or appear to behave) in line with developer intentions.

In practical terms, those concerned with Social Judiciousness may worry about recommender systems which maximize clickthrough rate but engender addiction or polarization (Stray et al., 2021; Agarwal et al., 2024; Prunkl, 2024; Smith et al., 2024), or worry that the values which get 'successfully' encoded in contemporary LLMs are instilled 'top-down' by large companies rather than having such val-

---

[3]Let $A$ denote the domain of 'acts', $Wa$ denote the fact that we – qua developers of the AI system – want some fixed AI system to do $a \in A$, and let $Da$ denote the situation where our trained AI in fact does or performs action $a$. If we formulate Positive Alignment as $\forall a \in A : \left(Da \rightarrow Wa\right)$ and Negative Alignment as $\forall a \in A : \left(\neg Wa \rightarrow \neg Da\right)$, then we can see that both express the same formula via contraposition.

[4]This was observed after the release of OpenAI's o3 and o4-mini models in comparison to GPT-4.5; see the PersonQA results present in (OpenAI, 2025b) and (OpenAI, 2025a), respectively.

ues decided via more democratic processes (Tang, 2025). These may very well be success cases for Task Reliability but failures of Social Judiciousness. Similarly, those concerned with Takeover Avoidance may worry that training LLMs to produce locally desirable outputs – for instance, by producing a CoT (chain of thought) that does not offend users or otherwise appear undesirable – may leave some undesirable behaviors intact, while increasing longer-term risks by causing a model "to hide its intent" (Baker et al., 2025).

## 3.3. Summarizing The Tradeoffs

This section has introduced two distinctions in order to highlight points of practical conflict in the pursuit of different alignment ideals. Because different conceptions of 'AI alignment' may be motivated by different threat-models or place primary focus on AI benefits or AI harms, we claim that disagreements about how to 'make AI systems more aligned' encompass competing technical priorities.[5]

| Alignment Ideal | Threat Model | Pos/Neg Alignment |
|---|---|---|
| Non-Takeover | Competence | Negative |
| Social | Either | Negative |
| Task | N/A | Positive |

*Table 2.* Alignment Ideals: Axes of Difference.

# 4. Recommendations for Future Practice

We now list several recommendations for future practice.

## Recommendation 1: Avoid Treating Distinctions of Policy As Distinctions of Scientific Scope

Consider claims that AI safety should focus on gradual disempowerment (Kulveit et al., 2025) or prioritize the future of work (Hazra et al., 2025). These are distinctions relating to *high-level policy ideals*. That is, 'gradual disempowerment' and 'the future of human work' pick out classes of societal outcomes which may be thought of or argued for as motivating ideals for technical researchers. In light of our discussion, we claim that researchers should carefully distinguish between high-level ideals of the kind cited and *model properties* we may want to target (or avoid) in near-term empirical work.

As noted by Kasirzadeh (2025), "advanced AI agents ... can be strategically leveraged for social engineering purposes such as ... amplifying existing prejudices, exploiting belief

---

[5]Threat-models were defined as a set of negative outcomes able to be addressed via interventions. We think it is most plausible to say that 'Task Reliability' is not founded upon threat-models per se, though we can imagine an argument for claiming that Task Reliability is primarily concerned with *Incompetence Harms*.

systems by synthetic evidence, and creating personalized epistemic bubbles" (2025, pg. 1987). Near-term work aimed at reducing the ability or propensity of AI systems to 'create personalized epistemic bubbles' (e.g., research attempting to mitigate LLM sycophancy) might thus be considered as an intervention which reduces the chance of gradual disempowerment. In turn, decreasing LLM sycophancy could require granting LLMs better access to 'ground truth' claims (thus enabling them to more easily push back on user framings), or reinforcing LLMs' tendencies to decline requests from users hoping for assistance in developing their pseudoscientific theories (see (Hill & Freedman, 2025) for discussion of a public case).

However, researchers concerned with 'gradual disempowerment' may equally well think that affording AI systems and their developers the ability to override the commitments of certain users could lead to an undesirable concentration of power. As such, researchers motivated by the ideal of 'gradual disempowerment' could, via better access to ground truth claims, be increasing LLMs' situational awareness (potentially harmful for *Takeover Avoidance*), or indeed be focused on developing 'non-judgmental' LLMs which avoid pushing back on potentially misguided user framings (potentially harmful for *Social Judiciousness*). While it is not our place here to proffer any advice concerning which high-level ideals to adopt, we think it important to distinguish between distinctions of *scientific or technical scoping* and distinctions between broader policy ideals or directions.

## Recommendation 2: Note Methodological Differences

Compared to our other two alignment ideals, Takeover Avoidance sounds fairly speculative. Indeed, while many prominent AI researchers treat Takeover Avoidance as important (Russell, 2019; Bengio, 2023; Milmo, 2024), other voices have variously judged such concerns as purely "hypothetical" (Domínguez Hernández et al., 2024), "ridiculously overblown" (LeCun, 2023), fantastically sci-fi (Ng, 2015), or "nonsense" (Bender, 2025). In some cases, researchers concerned with Social Judiciousness explicitly contrast 'concrete harms' with 'speculative risks', claiming the latter illegitimately "distract" from the former (Stark, 2024; Lahlou, 2025). For one camp, 'speculativeness' might be a necessary feature of conducting research concerned with the eventual trajectory of AI research, whereby temporary incompetencies should be expected to be resolved in the future. For another, theoretical and practical depth as applied to the *contemporary* technological landscape (as might be seen in invocations of, for example, 'the patriarchy' or 'racial capitalism') might be seen as necessary in order to surface harms which are otherwise opaque in ordinary technological practice. This methodological difference is likely to explain at least some portion of the disagreement between researchers who prioritize Takeover Avoidance and

those who don't.

However, we here note that differences in 'speculativeness' are differences of degree and not kind. To the extent that concerns of Social Judiciousness require theorizing about the societal effects of deploying AI systems, such theorizing requires 'more speculation' than analyses of AI performance with respect to more narrow technical criteria. Conflicts between these two ideals can be seen more or less explicitly in a 2022 analysis of papers published in the journals AIES and FAccT by Birhane and co-authors, which evaluated the degree to which each published paper "investigates, addresses, and/or mentions the disparate impacts of an algorithmic system" (Birhane et al., 2022, pg. 952), concluding with a call for future research to devote more attention to possible disparate impacts of AI systems.

Indeed, we note that methodological differences appear even when we restrict our attention to internal discussions of *Task Reliability*. Questions relating to the external validity of benchmarks, for example, are often questions about the degree to which our existing metrics capture more informal (and perhaps thicker) conceptions of 'capability'. Insofar as our more informal concepts of 'capable AI systems' outstrip our more formal and quantitative benchmarks, there exist gaps between 'our measurements of AI capabilities' and 'AI systems' *actual capabilities*', with assessments of the latter being more speculative than the former.

### Recommendation 3: Reviewer Pooling and The Diversity of 'AI Alignment'

'AI alignment' encompasses many distinct concepts. For this reason, we think it would be beneficial for: (i) researchers using the term 'AI alignment' to clearly identify the artifact 'being aligned' and target metric of 'alignment', and (ii) for major venues to distinguish alignment subareas in submission tracks and construct reviewer pools accordingly. The second suggestion in particular is well-supported by a 2025 paper investigating citation patterns for work on 'AI Ethics' (focused on 'Social Judiciousness') and 'AI Safety' (focused on 'Takeover Avoidance'), finding high degrees of citation homophily for both such communities (Roytburg & Miller, 2025). While our paper has focused on *conceptual* tensions between these ideals, empirical work provides evidence for *sociological clustering* — hence, instantiating distinct reviewer pools for these areas is likely to track distinct areas of expertise.

Our suggestions on this front nonetheless arrive with a notable degree of caution. While our paper has focused on three specific conceptions of 'AI alignment', we do not wish to suggest that the conceptions we introduce are the *only* such conceptions available. Some have called for a more 'bidirectional' approach to alignment between AIs and humans, while others have criticized the very concept

of 'alignment' as a process we impose onto AIs, thereby robbing us of the potential to learn from them (Agüera y Arcas, 2025). Indeed, if we view questions of AI alignment – very generally – as questions about what AIs *are* and *what we want to do with AI*, a yet more diverse range of conceptions emerge (Andreessen, 2023; Bostrom, 2024; Brynjolfsson, 2022; Danaher, 2019; Goldberg et al., 2024; Plurality Institute, 2025; Williams & Srnicek, 2017).

We thus do not wish to unduly reify the 'alignment' concepts discussed explicitly by this paper. Instead, we suggest that future discussions of 'AI alignment' either carefully specify their focus in light of existing ideals, or construct new alignment ideals in light of the distinctions outlined, or find new ways to carve up the space of concerns animating distinct alignment ideals.

### Recommendation 4: Use Qualified Alignment Terms

Consider Takeover Avoidance. Empirical researchers who aspire to produce work in service of this ideal do not, as of yet, have generally superhuman AI systems which they can try to 'align'. As noted in Section 2.3, researchers within this tradition sometimes talk about the 'goals' or 'preferences' of contemporary systems (see (Zhi-Xuan et al., 2025) for fuller discussion of this point). Hence, the *proxy alignment concept* used in such research may be thought of as a form of *Preference Alignment*, where (say):

> an AI system $S$ is 'aligned' if $S$: (i) possesses preference-like states, such that (ii) the preferences of $S$ are desirable preferences relative to some normative standard.

Of course, questions regarding what it means for an AI system to possess 'preference-like states' clearly remain. Nonetheless, using terms like Preference Alignment allows, in the first instance, those researchers sympathetic to the ideal of Takeover Avoidance to more easily discuss 'internal' questions regarding whether observed failures of Preference Alignment constitute meaningful progress towards the goal of Takeover Avoidance. Meanwhile, those less sympathetic to Takeover Avoidance can more easily separate criticisms of the observed results and their interpretation (e.g., 'should we treat the model as possessing preferences?', 'is the normative standard being used reasonable or well-specified?', etc.) from criticisms of the high-level ideal.

Likewise, our earlier discussion of Social Judiciousness distinguished between two possible sources of 'alignment' failure: Training Data Conservatism and Malicious Use. We think that future research in the tradition of Social Judiciousness should pay heed to this distinction, as different sources of harm lead to different routes for intervention. One unfortunate example to illustrate this point involves the absence of demographic diversity in the training data for

facial recognition technologies. For instance, facial recognition classifiers identified as racially biased (Buolamwini & Gebru, 2018) include those produced by IBM, which has tested facial recognition software within the NYPD (Allyn, 2020). To the extent one is concerned about such facial recognition technologies harming darker-skinned individuals through their use in predictive policing, discussions among researchers focused on Social Judiciousness – even among those sharing political ideals – ought to pay attention to the distinction outlined.

**Recommendation 5: Clearly Communicate Differences to Policy and Non-Technical Audiences**

Policymakers and scientific administrators are increasingly concerned with questions of 'AI alignment'. However, as our discussion has illustrated, different people may mean different things by 'AI alignment', and the term's polysemy in some cases leads to tradeoffs in practical priorities. Key decision-makers who remain unaware of these differences are thus less able to assess which priorities are most important for them, the different sources of evidence one might need in order to establish different sorts of concern, and the appropriate interventions to pursue given any such concern.

An example illustrating the importance of Recommendation 5 involves reference to the EU AI Act. The EU AI Act makes reference both to AI "alignment with human intent" (European Parliament, 2024, pg. 100) and to the importance of evaluating AI models using "appropriate technical tools and methodologies" (European Parliament, 2024, pg. 101). However, the act's discussion of auditing does not distinguish between the alignment ideals introduced by this paper; as such, policymakers mandating alignment audits lack clear and principled bases for distinguishing between the very different kinds of methodologies that may be required when auditing for (say) demographic biases and autonomous self-replication capabilities, respectively. Policymakers should thus be aware that these differences in terminology may have important consequences. Similarly, researchers across traditions should take care in distinguishing their own concerns from adjacent concerns, and communicating competing concerns in ways that illustrate the substantive issues raised by alternative camps even if one ultimately finds them unconvincing.

## 5. Alternative Views

We now defend our position against three alternative views.

### 5.1. Disagreements About 'Alignment' Aren't Disagreements About Technical Priorities

*Objection.* All disagreements concerning the referent of 'AI alignment' are best understood as either: (i) merely normative disagreements, or else; (ii) reflective of a single unified

concept of 'AI alignment' encompassing the concerns of every more specific ideal discussed in this paper.

*Reply.* The existence of a single unifying alignment ideal is (to put it baldly) simply implausible in light of our discussion. Indeed, part of the reason we have discussed tensions between different alignment ideals is to illustrate that such ideals are not merely concepts which are *compatibly different* (e.g., the concept 'red' and the concept 'square') but *incompatibly different* (e.g., the concept 'red all over' and the concept 'green all over').[6] Insofar as any intervention aids progress towards one such alignment ideal while vitiating progress towards another, there cannot be any single ideal which unifies them.

We also do not believe that disagreements between alignment ideals are purely normative. Almost no one wishes for an AI takeover, and so researchers preferring alternative alignment ideals over Takeover Avoidance must do so because they find the possibility of an AI takeover *implausible* — this is quite clearly an epistemic and not a normative difference. While disagreements about epistemic matters may themselves be intermingled with normative disagreements (e.g., regarding appropriate methodologies when thinking about the effects of AI systems), they remain disagreements about (even if speculative) matters of fact.

### 5.2. A More Modest Objection: 'Good Enough' Convergence

*Objection.* The previous objection was too strong. We grant that there exist *some* divergent priorities among different alignment ideals, but in practice such ideals are complementary and rarely conflict.

*Reply.* The second objection is more plausible than the first, and is occasionally hinted at in public talks (Dragan, 2025). And indeed we agree that there will exist shared technical projects which are positive by the lights of multiple distinct alignment ideals. For instance, techniques such as training data filtering may be useful from the perspective of both Takeover Avoidance and Social Judiciousness, while posing minimal costs to model capabilities that matter from the perspective of Task Reliability. Likewise, proposals such as third-party model evaluations aimed at improving robustness against harmful optimization — whether that be harms from *AI systems themselves* optimizing against human interests, the presence of harmful 'backdoors' implanted by unscrupulous AI developers, or the incautious rollout of narrowly capable but unreliable AI systems — may be similarly valuable from multiple perspectives.

---

[6] In slightly more outré language, one can read the claims of this paper as proposing that different working alignment ideals may stand in relations of 'determinate negation' to one another without standing in relations of 'formal' or 'abstract' negation (Hegel, 1807).

We nonetheless find the second objection implausible. Even the promisingly conciliatory case of model evals involves evaluations which "can vary widely in terms of method, approach, and subject matter", meaning that digital services coordinators in charge of overseeing external evaluations may be misled about which model properties it is most important to evaluate (Terzis et al., 2024). This in turn could lead to audits for models' 'autonomous self-replication ability' when one is most concerned with Social Judiciousness, or audits for models' demographic biases when one is most concerned with Takeover Avoidance. As we have highlighted the existence of *different* alignment ideals producing competing technical priorities *in at least some cases*, we believe that any claims of 'broad complementarity' between different alignment ideals require positive arguments that we have not yet found in the published literature.

### 5.3. The Paper's Claims Need Empirical Grounding

*Objection.* The paper aims to substantiate its views about alignment without reference to numbers on research trends, paper-keywords, citation analyses, interviews, or surveys. If 'alignment' really does encompass multiple competing technical ideals, this claim should be substantiated empirically.

*Reply.* Here we note a simple point of methodological disagreement. Before we can empirically study competing technical priorities behind different alignment ideals, we first need to understand the terrain conceptually: what *are* the underlying disagreements, such that we can evaluate whether a given empirical methodology successfully captures these differences?

Recall the earlier cited research from Roytburg and Miller, which found high degrees of citation homophily for researchers focused on 'Social Judiciousness' and those focused on 'Takeover Avoidance' (Roytburg & Miller, 2025). This analysis – while valuable, and while successfully providing evidence for the existence of *relatively insular* research communities – does not and cannot empirically address whether the ideals motivating such research are *in tension* with each other. It is for this reason that we believe that Roytburg and Miller's practical suggestions for 'unifying' these disparate literatures are premature, and use this juncture to note agreement with a prior position paper highlighting an absence of "conceptual clarity" as one pitfall of empirical ML research (Herrmann et al., 2024). With all that said, however, we believe that our paper provides value in part because it may help lay the conceptual groundwork for future empirical investigation of potential alignment tradeoffs. Consider the following research questions:

*RQ1.* At what rates do interventions reducing hallucination (a gain from the perspective of Social Judiciousness) measurably increase capabilities relevant to scheming or deceptive alignment (a concern from the perspective of Takeover Avoidance)?

*RQ2.* Does improving models' situational awareness (which may reduce certain incompetence harms) correlate with increased capacity for sandbagging on safety evaluations?

*RQ1* and *RQ2* are offered as potential research questions amenable to future empirical operationalization and investigation. Yet *RQ1* and *RQ2* are possible to formulate precisely only by way of disambiguating the multifarious concerns often conflated under the single heading of 'AI alignment'. Absent distinctions of the kind introduced by this paper, one cannot even *state* — let alone empirically investigate — research questions involving potential tradeoffs between Task Reliability, Social Judiciousness, and Takeover Avoidance.

## 6. Conclusion

The term 'AI alignment' encompasses multiple competing ideals. Sections 2 and 3 illustrated that 'AI alignment' was both polysemous, and that this polysemy resulted in *competing* and not merely *different* technical priorities. Together, these sections evidenced the paper's primary argumentative position. Thereafter, Section 4 expanded on the implications of our analysis. We suggested that researchers more clearly distinguish empirical 'alignment proxies' when discussing Takeover Avoidance, and more carefully distinguish between different sources of harm in cases of Social Judiciousness. More generally, we claimed that researchers would benefit from more explicitly specifying their background commitments without reifying the alignment concepts discussed herein.

We believe that the present paper provides value for reasons suggested in Section 5. We should try to understand disagreements about 'AI alignment' as they actually exist, which (for better or worse) do not come in the form of neatly packaged formalizations readily amenable to empirical study. In order to move towards more fruitful discussions about AI alignment, we need to acknowledge friction; we hope to have provided the rough ground.[7]

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
