# OpenReview forum: "Position: 'AI Alignment' Encompasses Competing Technical Priorities"
_ICML.cc/2026/Position_Paper_Track — ICML 2026 Position Paper Track regular_

### Official Review · Reviewer_KHaY · 2026-03-12

**Significance:** 3
**Argument Clarity:** 3
**Rating:** 4
**Confidence:** 3

**Questions:**

See Strengths and Weaknesses.

**Alternative Views Section:**

Yes

**Compliance With Llm Reviewing Policy A Conservative:**

Affirmed.

**Discussion Potential:**

3

**Final Justification:**

My main concern is addressed, and I think the strengths of the paper outweigh the weaknesses. So, I will keep my positive rating for the paper.

**Paper Summary:**

This paper argues that the term “AI alignment” encompasses multiple competing technical ideals, not just differing goals. The authors identify three primary conceptions of alignment: task reliability (alignment with developer intentions), social alignment (alignment with broader societal values), and takeover avoidance (alignment aimed at preventing catastrophic AI takeover). The authors show that these ideals can conflict in practice, e.g., improving AI competence may reduce social harms but increase takeover risks. They recommend using more precise terminology, distinguishing between sources of harm, and communicating these differences clearly to policymakers.

**Position:**

Yes

**Position In Title:**

Yes

**Related Work:**

3

**Strengths And Weaknesses:**

**Strengths**

- The paper offers a well-structured taxonomy of alignment concepts, clarifying how different research communities use the term “alignment” in different ways. This helps reduce misunderstanding and enables more targeted discourse.

- The paper provides useful guidance for researchers and policymakers navigating real-world alignment decisions by identifying concrete trade-offs between alignment ideals (e.g., competence vs. safety).

**Comments**

- The paper acknowledges a lack of empirical grounding for its claims about competing priorities, relying instead on conceptual analysis, which limits the strength of the conclusions about how often such trade-offs occur in practice.

- While the paper highlights tensions between alignment ideals, it offers relatively little discussion of how to resolve or manage these trade-offs in technical or policy settings.

**Support:**

3

---

> ### Author Rebuttal · Authors · 2026-03-31
>
> We thank the reviewer for their attention to feedback on the paper, and now address two points of concern raised in the review.
>
> **Empirical Evidence**. First, our "lack of empirical grounding" is said to limit "the strength of the conclusions about how often such trade-offs occur in practice." This concern is addressed directly in Section 5.3, where we argue that conceptual clarification is a necessary precursor to empirical study. Before we can measure how frequently polysemous alignment ideals lead to confusion in practice, we first need to clearly state what the underlying disagreements are. We also note that the ICML position paper track is explicitly designed to allow for non-empirical contributions. Indeed, an ICML Position Paper from 2024 (”Why We Must Rethink Empirical Research in Machine Learning”) identifies a ”lack of conceptual clarity” as one pitfall of empirical ML research. Our revision will now make this argument explicitly in Section 5.3, and we hope that a later mentioned revision in Section 5.2 (see our response to ‘Managing Tradeoffs’) on model auditing helps satisfy the reviewer on this front.
>
> Our contribution should be read as laying the conceptual groundwork for more substantive empirical investigation of these tradeoffs in the future. Indeed, our framework generates specific research questions which -- while beyond the scope of a position paper -- require precisely the kind of conceptual scaffolding we provide. Consider:
>
> 1. “At what rates do interventions reducing hallucination (a gain from the perspective of Social Alignment) measurably increase capabilities relevant to scheming or deceptive alignment (a concern from the perspective of Takeover Avoidance)?”
>
> 2. “Does improving situational awareness in models -- which may reduce certain incompetence harms -- correlate with increased capacity for sandbagging on safety evaluations?”
>
> Questions (1) and (2) are illustrative, and we think amenable for future empirical operationalization and investigation. Yet it is only possible to formulate precisely only using distinctions of this kind we introduce. We take this to be a meaningful contribution, and our revisions now highlight this contribution more explicitly.
>
> **Managing Tradeoffs**. Second, it is claimed we offer "relatively little discussion of how to resolve or manage tradeoffs in technical or policy settings". There are parts of this diagnosis with which we agree and parts with which we disagree. On the one hand, we agree that our paper does not ‘resolve’ these tradeoffs. We do not aim to unilaterally propose resolutions across diverse areas of research, and instead aim to provide the conceptual infrastructure that will enable researchers to bring their expertise from different subdisciplines to move towards more well-informed constructive resolutions.
>
> While our list of recommendations offered in Section 4 provides an initial set of management mechanisms for how to more productively navigate these tradeoffs, we agree that more detail could be provided and appreciate the helpful suggestions regarding the benefits of our contributions for research and policy. To that end, our revision will do two things.
>
> In the first instance, we will expand our discussion of model evals (referencing Terzis et al 2024) in Section 5.2 to better highlight practical relevance; specifically, we will note that the EU AI Act’s discussion of auditing does not distinguish between the alignment ideals outlined in the paper. Indeed, we note that -- absent distinctions of the kind we provide -- a policymaker mandating ‘alignment audits’ has no principled basis for deciding whether to audit for demographic bias or autonomous self-replication capability despite these requiring very different evaluation methodologies.
>
> In the second instance, we will explicitly suggest that: (i) researchers clearly identify the artifact 'being aligned' and target metric of ‘alignment’ (see our Q1-Q2 decomposition), and; (ii) that major venues distinguish alignment subareas in submission tracks and construct reviewer pools accordingly. The benefits of our second suggestion are well-supported by the citation homophily evidence referenced in Section 5.3.

---

> > ### Author Rebuttal · Reviewer_KHaY · 2026-04-01
> >
> > The authors have provided a detailed response which has addressed my concerns.

---

### Official Review · Reviewer_WVnK · 2026-03-13

**Significance:** 2
**Argument Clarity:** 3
**Rating:** 5
**Confidence:** 4

**Questions:**

I don't have a lot of additional questions here (my suggestions for how to improve the paper are included above in the strengths and weaknesses section), but I will ask, to help frame the paper, what was the motivation for developing this set of definitions? Perhaps that can help inform a stronger case for their usefulness.

Also, as a minor nitpick, please fix the line wrap on some URLs in the references. It's not a big deal but it makes a couple of the references hard to read due to overlapping text.

**Alternative Views Section:**

Yes

**Compliance With Llm Reviewing Policy A Conservative:**

Affirmed.

**Discussion Potential:**

4

**Final Justification:**

The authors elaborated their intended position, which is well supported by the existing manuscript's arguments and just needs a little reframing to presented more directly (it was already present in the paper, but not clearly framed as "the point" in my reading). As I believe this is in scope for revision here and reasonably achievable, I've raised my score.

**Paper Summary:**

This paper outlines how the concept of "alignment" in AI research refers to multiple distinct goals (minimally 3) which should be acknowledged and treated distinctly in research and communications both within and without the field.

**Position:**

Yes

**Position In Title:**

Yes

**Related Work:**

4

**Strengths And Weaknesses:**

Overall, this paper is excellently written and persuasive. It examines and dissects the concept of "alignment" in ways both specific and comprehensible, with illustrative examples throughout as well as example citations for the patterns discussed. That said, there are some passages which suggest insufficient proofreading and the need for another pass- for example, in section 3.1 there is an empty "()" which is intended to enclose a citation that is not present in the paper. Grammar issues and excess repetition are also scattered intermittently throughout the paper, and while it doesn't rise to a level that impedes comprehension, such issues should be cleaned up.

Beyond that, my core issue with this work is simple: while I appreciate the insight from examining the concept of alignment this way, the paper isn't really asserting a position- it simply presents a set of (thoughtful and useful) definitions that are useful when thinking and talking about alignment research. While I can see how someone might find this act of defining alignment this way to be itself a position, in my opinion (and I'm open to persuasion here) these definitions are logically self-evident, being definitions which don't require justification beyond a clear and specific assertion. The paper does provide some recommendations to use these definitions in communication, but this seems very much secondary and not the focus of the paper (as written).

I think to really shine this paper needs to make a stronger case for why defining the field of alignment using the proposed terms will benefit research or policy or other factors of interest to the audience, or what should be done based on this distinction into three separate topics. For example, arguing that categorizing all three topics as "alignment" hurts progress/public reception/etc for one or more of the types of alignment, and that researchers ought to insist on identifying their work more specifically, to the point of calling on conferences to apply different labels and different reviewer pools for each. I can see a number of similar positions that could be taken, but I think it's key to make a stronger case for why this thoughtful and well explained set of definitions is useful/important.

Given all of that, I am torn, but I think marginally inclined to recommend rejection on the grounds that this paper could be made much stronger and higher impact and resubmitted (hopefully NeurIPS runs the position paper track again this year). I think this work could be very impactful and promote valuable discussion and a change of perspective in the alignment community, but it needs a clearer motivation for why the community should sit up and take notice to really land.

**Support:**

4

---

> ### Author Rebuttal · Authors · 2026-03-31
>
> We are grateful for the careful engagement and constructive suggestions. To begin, we have cleaned up typesetting and grammatical issues alongside the empty citation. We now offer two points of response and discuss how we aim to address them.
>
> **Our Taxonomy and Position**. While our taxonomy plays an important argumentative role, the primary position is not the taxonomy itself. Rather, our position is that the term 'AI alignment' encompasses competing technical priorities. The definitions offered in Section 2 should be read as the apparatus needed to make the competition claim precise rather than ends in themselves.
>
> To that end we substantiate our position primarily in Section 3. For example, Section 3.1 argues that reducing LLM hallucination rates may aid progress towards the ideal of Social Alignment while aggravating the difficulty of Takeover Avoidance. Likewise, 3.2 mentions tradeoffs between Task Reliability and the remaining ideals. Without distinguishing Task Reliability, Social Alignment, and Takeover Avoidance, one cannot even *state* that reducing hallucination rates is simultaneously progress on one alignment ideal and regress on another. Thus we agree with the review’s points about the justificatory role of definitions, though devote significant attention to constructing definitions because the polysemy of ‘AI alignment’ currently occludes these important differences.
>
> For this reason we believe that our primary claim — namely, that common alignment ideals encompass competing technical priorities — is a position in the sense relevant for ICML. Importantly, we also take our position to be importantly distinct from (e.g) more prosaic claims about resource conflicts, where research furthering 'AI alignment' in one sense may compete for resources (such as research attention or policy prioritization) with 'AI alignment research' in another sense. We think that such claims (while true) would be more trivial. Instead, we aim to highlight that ‘AI alignment’ encompasses specifically technical priorities which are competing rather than merely different.
>
> **Our Work’s Significance**. Given increased attention to questions of AI alignment in both technical and policy research, we think conceptual clarity regarding 'AI alignment' is important. Without a more precise sense of what we (and those in different traditions) mean by 'AI alignment', it is likely that we will miscommunicate with one another, and create infelicitous environments for policymakers trying to understand and regulate AI. Absent distinctions of the kind we provide, a policymaker mandating ‘alignment audits’ has no principled basis for deciding whether to audit for demographic bias or autonomous self-replication capability despite these requiring very different evaluation methodologies.
>
> We would also like to respond to the question regarding our motivation. Although our work is conceptual rather than mathematical, we view our motivation as in one way analogous to Kleinberg, Mullainathan, and Raghavan’s 2016 work illustrating tradeoffs between different formal notions of 'fairness’. Highlighting tradeoffs between different notions of fairness allowed us to better understand where different fairness ideals conflict, and aim for different fairness ideals in different contexts. Of course, we do not claim a formal impossibility result (indeed, the ‘fuzziness’ of AI alignment noted by Reviewer amDH makes this difficult). Still, we believe the structural parallel helps illustrate both our motivation and the nature of our contribution.
>
> **Minor Edits**. We shall incorporate many helpful suggestions raised by this review. One suggestion mentioned that conferences could apply different labels and reviewer pools to each alignment ideal, and we agree this strengthens the paper. As such we will expand Recs 3 and 4 by specifically suggesting that major venues distinguish alignment subareas in submission tracks, and safety evaluations explicitly state their operative alignment ideal. On this point, we also wish to note that Section 4’s recommendations are not merely incidental to our position. If common alignment ideals encompass competing technical priorities, then concrete consequences follow for how researchers specify their operative alignment ideal (Rec 3), how policymakers design eval regimes (Rec 4), and how the field acknowledges methodological differences across alignment traditions (Rec 1).
>
> The general points raised about practical upshots are also well-taken. To that end, we will expand our discussion of ‘model evals’ (referencing Terzis et al 2024) in Section 5.2 to better highlight its practical relevance. Specifically, we will make explicit reference to the EU AI Act’s discussion of model auditing, note that it does not distinguish between the alignment ideals outlined in the paper, and elaborate why model evaluations aimed at (for example) Social Alignment may lead to undesirable consequences from the perspective of Takeover Avoidance.

---

> > ### Author Rebuttal · Reviewer_WVnK · 2026-04-04
> >
> > I like this explanation- thanks for clarifying, it helps a lot. With a little tweaking of the introduction to put this argument front and center and establish it as a non-trivial position as in this rebuttal (which seems in scope to revise here), I think this is a good paper and will raise my score.

---

### Official Review · Reviewer_amDH · 2026-03-13

**Significance:** 4
**Argument Clarity:** 3
**Rating:** 6
**Confidence:** 3

**Questions:**

see my previous comments (weakness section)

**Alternative Views Section:**

Yes

**Compliance With Llm Reviewing Policy A Conservative:**

Affirmed.

**Discussion Potential:**

4

**Final Justification:**

I believe the authors are asising important and valuable questions with their position paper. I have raised my score accordingly.

**Paper Summary:**

This position paper discusses the problem of AI alignment. The position of the paper is that 'AI alignement' is polysemous, as it encompasses different notions about the objects of alignment, or about the threat models they undergo. More specifically, they distinguish between three types of alignment: task-level (the AI system performs the task it is programmed to), social (the AI system is aligned with some social norms), or takeover avoidance (while optimizing a given goal, an AI system can cause harms or have negative outcomes).

Interestingly, the authors show that those alignment ideals can be generally competing, depending on the nature of the threat they are subject to. They distinguish between harms from (in)competence, and competition between negative and positive alignments (maximize benefits versus minimize harms).

The recommend, as a call for actions, to 1) acknowledge methodological differences, and pay more attention to disparate impacts of AI systems; 2) acknowledge the diversity of notions of alignment; 3) use qualified alignment terms and 4) have a better communication toward policymakers and non-technical audiences  on this topic.

The alternative views gather 3 points: 1) the underlying technical priorities are the same, whichever the sens in which we understand alignment; 2) alignment ideals are complementary and not (in general) competing and 3) the conceptual works on this paper should require empirical grounding.

**Position:**

Yes

**Position In Title:**

Yes

**Related Work:**

3

**Strengths And Weaknesses:**

strengths:
- the positioning of this paper is much relevant to the community, and frame efficiently the different concepts and challenges behind this fuzzy notion of AI alignment. It has a strong potential to trigger discussions among researches concerned by alignment but not only. As such, its importance is deemed high by the reviewer.
- this position paper is nicely written (apart for some typos that I noted at the end) and clear.
- the positioning wrt. related works seem good to me, though I acknowledge not to be an expert in this domain.

Weaknesses/suggestion of improvements:
 - beyond the notion of societal, normative alignment, a notion of economical or environmentally sustainable alignment could be evoked as well (i.e. an AI system can be aligned with cost objectives or maximal CO2 emission rate). Maybe this falls down under the concepts of social alignment).
 - this is more of a commentary rather than a weakness: I wonder to some extent wether this conceptual discussion about what is aligned and the associated tradeoff should be totally disconnected from the technical means to perform this alignment. A paragraph, or even a column, about how algorithmically this alignment is performed (RLHF, guardrails, dataset curations, etc.) and also how it can be measured (which is totally absent from the paper) could have highlighted the interconnections between the 'what' and the 'how'. Notably,  the state of what is possible as of today (even though we do not dispose of AGI for now) is unclear to me.

Typos:
l 153 -> Compatability -> Compatibility
l 130, c2 - > ICG undefined
l 160, c2 - > Authors mention 4 ideals but it is only 3
l 185, c2 -> Missing reference after ‘facial recognition’
l 195, c2 -> repatition ‘reduce the tendency’
l 216, c2 -> one of the two ‘Competence’ should be ‘incompetence’
l 228 -> what is FOL ?
Table 2 -> it is not really clear what ‘Either’ or ‘Neither’ refer to

**Support:**

3

---

> ### Author Rebuttal · Authors · 2026-03-31
>
> We thank the reviewer both for their feedback and for their nice summary of our paper. In light of the suggestions we have revised the paper to correct all typos, clarified ambiguities regarding 'FOL’ (first order logic), and made the labelling of Table 2 by adding clarificatory detail in the main text of Section 3.3 before Table 2 is introduced.
>
> The point raised about environmental metrics and cost objectives is interesting. We are tempted to say that concerns of this kind would fall into two camps. First (as the reviewer themselves suggests) many cases of this kind would be read as concerns of 'Social Alignment’, insofar as the deployment of models with high CO2 emissions exacerbates undesirable trends. Second, we think at least some cases would be properly considered out of scope for our analysis. For instance, we are not aware of any community using 'AI misalignment’ to refer to cases whereby AI systems are declared 'misaligned’ in virtue of the cost required to develop it. Although we agree that an extended conversation on a wider range of normative demands upon AI systems should also take into account labor and environmental costs of developing them, we will add a footnote to clarify the importance of such costs even though we consider them out of scope for our present analysis.
>
> The reviewer also notes the following two points: (i) ideally, conceptual discussions about 'AI alignment’ would be integrated into technical discussions of how to 'align’ such systems (however conceived). And: (ii) that greater clarity on how to measure progress towards different alignment ideals would be beneficial.
>
> Here we have substantial agreement with the reviewer. In part, we omit technical discussions of how to align such systems and how we might measure ‘alignment’ (however conceived) due to constraints of space and scope. Many researchers motivated by Takeover Avoidance (or Social Alignment, or Task Reliability) disagree substantially on which methods are effective, and indeed the methods used may vary considerably depending on the type of system one wishes to 'align’. Likewise, yet further disagreements exist on issues of how one ought to measure progress towards the ideal. While the specific methods used by researchers may be plural, they are often evaluated by the community on some shared standards — and our position argues that those shared standards require disambiguation. But we agree that adding some illustrative examples can illuminate this point better. While an exhaustive survey of such techniques would clearly exceed the paper’s budget, our revisions will add some short content noting RLHF and training data attribution methods as illustrative examples.
>
> Overall, though, we share the reviewer’s wish for greater integration between technical and conceptual discussion. Our hope is that, by offering more explicit clarity about various different alignment ideals, future technical work is able to more fluidly integrate more conceptual research into technical discussions of measurement and training techniques.

---

> > ### Author Rebuttal · Reviewer_amDH · 2026-04-01
> >
> > Thank you for the clarifications and the discussion. After reading the other reviews, I still believe this paper is a good fit for this ICML position papers track. I am raising my score accordingly.

---

### Official Review · Reviewer_53Ly · 2026-03-16

**Significance:** 4
**Argument Clarity:** 3
**Rating:** 5
**Confidence:** 4

**Questions:**

The paper mentions the application of AI in fields such as healthcare and finance, but does not analyze the industry compatibility. May I ask if there is a plan to conduct a scenario-based study on alignment priorities? Will there be the proposal of differentiated alignment norms and practical paths for different industries?

**Alternative Views Section:**

Yes

**Compliance With Llm Reviewing Policy A Conservative:**

Affirmed.

**Discussion Potential:**

3

**Ethics Review Area:**

["Legal Compliance (e.g., EU AI Act, GDPR, copyright, terms of use)", "Research Integrity Issues (e.g., plagiarism, collusion rings, etc.)"]

**Final Justification:**

My concerns have been addressed. I decide to maintain my current score.

**Paper Summary:**

In recent years, "AI alignment" has become a core topic in the field of artificial intelligence. The paper systematically examines the core differences among various "AI alignment" concepts, reveals the underlying conflicts in technical priorities, and proposes research questions that align with the current practical needs of AI alignment research, thus having significant academic clarification and practical guidance significance.

**Position:**

Yes

**Position In Title:**

Yes

**Related Work:**

3

**Strengths And Weaknesses:**

1. The paper addresses the three core questions of "What is the alignment object (x)", "To what target is it aligned (y)", and "What attributes must be met (P)", thereby decomposing "AI alignment" into three core ideals: task reliability, social alignment, and takeover avoidance. Through a table, it clearly defines the core characteristics of each ideal, with clear concepts and distinct boundaries.
2. The paper mainly focuses on the three core ideals of "task reliability, social alignment, and takeover avoidance", but in recent years, the academic community has proposed many new alignment concepts, such as Decolonial Alignment, Socioaffective Alignment, and Collective Alignment. The paper only briefly mentions these emerging concepts in the introduction, without analyzing their core connotations, conflicts with traditional ideals, and practical approaches, resulting in limited coverage of the research.
3. The paper mainly focuses on the conflict of technological priorities, but the practical advancement of "AI alignment" is also influenced by many non-technical factors, such as policy regulatory orientation, industry interest demands, public perception and acceptance, etc. For example, policymakers may prioritize "social alignment" due to short-term social stability needs, while technology companies may focus on "task reliability" due to commercial interests.

**Support:**

3

---

> ### Author Rebuttal · Authors · 2026-03-31
>
> The attention to and kind remarks about our paper are appreciated. We are particularly pleased that the paper was seen to provide “significant academic clarification and practical guidance”.
>
> Two points of potential weakness are raised in this review. First, it is noted that we reference many specific alignment ideals (Socioaffective Alignment, Decolonial Alignment, etc.) in the introduction without analyzing their core connotations. Here, we agree that our coverage of the research is non-exhaustive. As it turns out, we happen to believe that the various alignment concepts cited in the introduction can be incorporated into our taxonomy. For instance, we later reference Foster’s concept of 'Thick Alignment’ when discussing Social Alignment (in Section 2.2), and note here that the referenced concept of 'Decolonial Alignment’ can be understood to be within the purview of Definition 2.2. In the case of Decolonial Alignment, the “undesirable trends” referenced in our definition of Social Alignment involve the imposition of colonial and hegemonic categories.
>
> That said, we do acknowledge that (among other things) constraints of space necessitate less than maximal coverage of all research falling under the name of 'AI alignment’. Indeed, we think there is an argument for claiming that socioaffective and collective notions capture a potential fourth category relating to how systems may enhance overall sociotechnical capabilities when integrated into human systems. However, concepts of Socioaffective and Collective Alignment are still very much nascent within the broader literature which to some extent fetters our ability to provide a careful definitional treatment. Whether such concepts introduce genuinely novel Q1-Q2 pairings or represent important elaborations within existing ideals remains an open question we hope future work will address. In Section 4 we explicitly caution against “reifying” the alignment concepts offered by our paper (see Recommendation 2) in part because we wish to understand the nature of existing discussion better rather than calcifying its current shape.
>
> Second, the reviewer notes that our paper '”mainly focuses on the conflict of technological priorities”, even though “many non-technical factors” are important for practically advancing ideals of AI alignment. We agree that non-technical factors are important, and value research that aims to explicitly examine the interplay of technical considerations with social and economic incentives. While our focus in this paper is more tightly constrained, we consider our work complementary to such initiatives. Indeed, we believe that our paper can help inform academic and public perception between stated and de facto priorities. By understanding (for example) the distinction Task Reliability and Social Alignment, it becomes easier to notice when commercial interests may favor one without serving the other. This is because our paper now provides a vocabulary for reasoning through tradeoffs and (where appropriate) thinking about concepts of accountability which might otherwise be obscured by the polysemy of ‘AI alignment’.
>
> Finally, the reviewer also asks if we plan to conduct a scenario-based study on alignment priorities. We do not currently plan to conduct such a study ourselves, though we would be interested to see other researchers’ attempts to propose differentiated alignment norms along the lines suggested.

---

> > ### Author Rebuttal · Reviewer_53Ly · 2026-04-03
> >
> > Thank the authors for their responses. My concerns have been addressed. I decide to maintain my current score.

---

### Decision · Program_Chairs · 2026-04-30

**Decision:**

Accept (regular)

**Comment:**

Multiple reviewers mentioned the paper’s clarity, strong framing/concept, identification of concrete trade-offs between different alignment objectives, and high potential to stimulate discussion as its key strengths. There were some concerns that the analysis was largely conceptual, lacking empirical grounding of some aspects, and leaving open the question of how often these trade-offs arise in practice and how significant they are when they do. The rebuttal did address some of these concerns by clarifying hte central claim, expanding the discussion of applications (e.g, policy audits/evaluation), although some gaps in empirical grounding still remain. In the subsequent revision, the authors are advised to improve the depth of the grounding/analysis as much as possible, incorporating aspects of your rebuttal, and address any remaining concerns raised in the reviews. The reviewers generally agree that the strengths of this paper outweigh its few remaining weaknesses.